# Dynamic Expansion of Urban Land in China's Coastal Zone since 2000

**Peipei Du** [1,2,3,4], **Xiyong Hou** [1,3,4,*] **and He Xu** [1,2,3,4]

1  Yantai Institute of Coastal Zone Research, Chinese Academy of Sciences, Yantai 264003, China;
   ppdu@yic.ac.cn (P.D.); hxu@yic.ac.cn (H.X.)
2  University of Chinese Academy of Sciences, Beijing 100049, China
3  CAS Key Laboratory of Coastal Environmental Processes and Ecological Remediation,
   Yantai Institute of Coastal Zone Research, Chinese Academy of Sciences, Yantai 264003, China
4  Shandong Key Laboratory of Coastal Environmental Processes, Yantai Institute of Coastal Zone Research,
   Chinese Academy of Sciences, Yantai 264003, China
*  Correspondence: xyhou@yic.ac.cn; Tel.: +86-0535-210-9196

**Abstract:** Although a major region with strong urbanization, there is not yet a systematic and comprehensive understanding of urban expansion during the last 20 years for China's coastal zone. In this paper, based on remote sensing techniques, and using indicators such as new urban land proportion, annual urban increase, and annual growth rate, as well as a landscape expansion index reflecting the urban expansion type (e.g., edge-expansion, infilling, and outlying), we measured the dynamic expansion of urban land in China's coastal zone since 2000. The results indicated that: (1) China's coastal zone experienced rapid urbanization from 2000 to 2020, with the new urban land and annual urban growth rate at 17,979.72 km$^2$ and 4.83%, respectively. The new urban land was mainly concentrated in economically advanced regions, such as Bohai Rim, Shandong Peninsula, the Yangtze River delta, and the Pearl River delta. (2) The urban growth rates of coastal cities in Liaoning, Hebei, Shandong, southeast Fujian, and Taiwan became slower over time, with a sharp decline during 2015–2020. In the mid and south of China's coastal zone, such as coastal cities in Jiangsu, Guangxi, and Hainan, there was slow urbanization before 2015, and urban land expanded dramatically during 2015–2020. (3) The urban expansion of China's coastal zone was dominated by edge-expansion after 2000, but it went through a low-speed and intensive development stage during 2010–2015, with an increase in urban land less than 50% of that in the other three five-year periods, and the most significant filling of urban space compared with the other three five-year periods, which was probably caused by the global financial crisis. (4) The spatial-temporal differences in the urbanization process in China's coastal zone were largely consequent on national economic development strategies and regional development plans implemented in China's coastal zone.

**Keywords:** coastal area of China; urban expansion; spatial-temporal difference



## 1. Introduction

Urbanization is the result of a combination of economics, politics, and technology [1,2]. With urbanization, more than half of the world's population live in urban areas, and the proportion will reach 68% by 2050, with the strongest increases in developing countries [3,4]. Rapid urbanization has led not only to a dramatic increase in urban populations, but has also brought a series of problems, including pollution [5–7], shortage of resources [8,9], decrease in biodiversity [10–12], and so on, which have seriously affected the local climate, ecological environment and global change [13]. It is well known that urban expansion and population expansion occur at different rates; it is expected that by 2030 the speed of global urban land expansion will be three times faster than that of population expansion [14,15]. The rapid expansion of urban land has led to the loss and abandonment of farmland, reduction in forest cover and other ecological damage. Analyzing the spatiotemporal

characteristics and dynamic expansion of urban land is important for understanding the ecological effects of urbanization and optimizing urban land use patterns [16].

Remote sensing is considered a major method for studying urbanization—it can accurately extract urban land use and cover (LUCC) information based on spectral features and structural characteristics of land features and monitor the dynamic changes of urban land over multi-temporal scales. Currently, with advances in remote sensing technologies and methods, many regional and global LUCC maps or urban distribution maps have been produced [17–19], such as LandScan, the Global Rural-Urban Mapping Project (GRUMP) and GlobeLand30. These products provide useful information on urban extent for monitoring the spatial characteristics and dynamic changes of urban land, and numerous studies have been performed to analyze urban expansion at regional, national or global scales [20,21].

In addition, quantitative indicators of urban expansion and landscape metrics are often used to describe the temporal-spatial characteristics of urban expansion [22–25]. Indicators, such as increase, speed, and intensity, can help quantify and compare the urban expansion of cities or regions at different stages. Moreover, landscape metrics, such as shape, density, and aggregation, can identify the urban expansion forms or types in different dimensions. Yang et al. (2019) [26] used three quantitative indicators (i.e., expansion rate, annual increase, and annual growth rate) and seven landscape metrics (i.e., landscape expansion index, fragmentation index, patch density, number of patches, landscape shape index, largest patch index, and percentage of landscape) to analyze the spatiotemporal evolution of urban agglomerations. In sum, the combined application of multiple methods, including spatial analysis techniques, multi-temporal land use/cover data, measurable indicators and landscape metrics, is necessary to measure spatiotemporal characteristics and dynamic expansion of urban land for a certain region or cities over time.

Numerous studies have been performed to investigate the dynamic evolution of urban land, involving a single city, several cities, urban agglomerations, provincial capitals, and cities or agglomerations for comparison [16,24–32]. Developed countries, such as the United States and Japan, have been highly urbanized since the 1970s, and current studies on the dynamic evolution of urban land have mainly focused on a single city, or urban agglomerations with a number of cities, such as Kentucky [27], the Tokyo metropolitan area [28], and so on. For developing countries, such as India and China, many studies have focused on large cities, urban agglomerations, and typical cities, while only a few studies have paid attention to small cities. For example, Shukla and Jain [29] quantified the urban growth trajectories and spatiotemporal pattern of Lucknow in India based on LUCC classification maps covering the years 1990, 1999, 2009, and 2016. They found that the urban core and suburban areas became more aggregated and diffuse, respectively. Many scholars have pointed out that China's urban development has suffered from disequilibrium [16,30,31], with excessive urban expansion, unreasonable spatial forms, and has often been accompanied by certain kinds of urban disease, such as ghost cities, traffic jams, urban heat islands, and urban water logging, etc., thus hindering the sustainable development of a single city or cities in a certain region.

It is well known that China has experienced an unprecedented urbanization process with amazing speed and at massive scale since the implementation of the "reform and opening-up" policy [31,32]; the urbanization rate in China increased from 17.9% in 1978 to 60.6% in 2019 [33]. As the world's largest developing country, China's urbanization process has attracted worldwide attention. The urbanization process in China is influenced by many factors, such as resource endowment, population concentration, and policy orientation. Cities with high administrative rank, abundant resources and superior geographical location are able to gain more development opportunities. The differences in these factors have led to significant regional differences in China's urbanization process, which are reflected in the area of new urban land, the pattern of urban expansion, and the rate of urban growth. In general, larger cities have more newly expanded urban land. Li et al. (2015) [34] proposed that urban expansion in China was sensitive to a city's rank based on the analysis of statistical data. Liu et al. (2021) [31] used an RS method to analyze the urban

expansion of 75 typical cities in China from the 1970s to 2020. This suggested that China's urbanization process was strongly correlated with the administrative rank and population size of cities. Sun and Zhao (2018) [32] found that the urban growth rate was inversely proportional to city size by analyzing 13 cities across the Jing-Jin-Ji urban agglomeration, which was consistent with the results of Zhao et al. (2015) [35]. However, urbanization is a dynamic evolutionary process with significant spatial differences [36], such that large cities with a continued increase in urban land might exhibit a downward trend in urban growth rate compared to small cities [26,32]. China has many cities of different scale and it is of value to research the dynamic changes in urban land as well as the relationship between city scale and urban growth rate to meet the needs of future urban planning and management.

As a region with advanced economic development, China's coastal zone spans a large area from north to south, with climatic and geographical differences leading to an extremely uneven distribution of the population along the coast [37,38]. The urbanization process in China's coastal zone is significantly affected by resource distribution and the impacts of national and regional development policies [39,40]. A number of studies have been carried out to monitor the spatiotemporal characters of urban land in China's coastal zone [24,39,41–44]. Most of these have attached importance to large and developed cities, such as Guangzhou [24], and urban agglomerations [23,39]. However, little attention has been paid to small and medium-sized cities in China's coastal zone, although they constitute a higher proportion than large cities, and research studies focused on urban expansion of all the cities in China's coastal zone are even rarer [16]. Therefore, it is necessary to perform more research on urban expansion for the entire coastal zone in China.

In this paper, we quantified the spatiotemporal characteristics of urban expansion in China's coastal zone over the past 20 years using multi-temporal land use data for 2000, 2005, 2010, 2015, and 2020 [45]. This research aimed to (1) quantify and compare urban extent, new urban increase, urban growth rates, and urban expansion types in cities of China's coastal zone during different periods; (2) analyze the spatiotemporal differences in urban expansion at city level and regional level; and (3) discuss the consistencies between social-economic development policies and urban expansion in the cities of China's coastal zone.

## 2. Materials and Methods

### 2.1. Study Area

China's coastal zone is located at the intersection of Eurasia and the Pacific Ocean, with a continental coastline of more than 18,000 km and an island coastline of more than 14,000 km, including Liaoning, Hebei, Tianjin, Shandong, Jiangsu, Shanghai, Zhejiang, Fujian, Guangdong, Guangxi, Hainan, Hongkong, Macao and Taiwan (Figure 1). Over 50% of China's large cities, more than 40% of its population and 60% of its GDP, are concentrated in the coastal zone [46]. In mainland China, coastal prefecture-level cities and some prefecture-level cities that are closer, but not adjacent, to the continental coastline, are defined as the coastal zone in this paper; in particular, Taiwan island and Hainan island are both included in this study, with Taiwan as a provincial level and Hainan dividing into Haikou, Sanya and other cities.

### 2.2. Land Use Data

The land use data of 2000, 2005, 2010, and 2015 were obtained from the China Coastal Land Use Database [45,47], which is produced by a visual interpretation method based on multi-temporal Landsat images. This database divides land use types into 8 primary categories and 24 secondary categories (Figure 2); the primary categories include farmland, forest land, grassland, built-up area, inland freshwater, coastal saltwater, human made wetlands, and unused land. Subsequently, we updated the land use database based on Landsat/OLI images of 2020 according to the method provided by Di et al. [45]. Thus, complete land use data covering 2000, 2005, 2010, 2015, and 2020 were accessed. The land use data for 2010, 2015, and 2020 were selected to evaluate mapping accuracy; in particular,

validation samples were collected from high resolution images on Google Earth, and the overall accuracy and Kappa coefficient were calculated for the 8 primary categories of land use. The results (as shown in Figure 2) confirmed that high accuracy of land use mapping was achieved. In this paper, urban land information was derived from the secondary category denoted 'city' and numbered 41 in Figure 2.

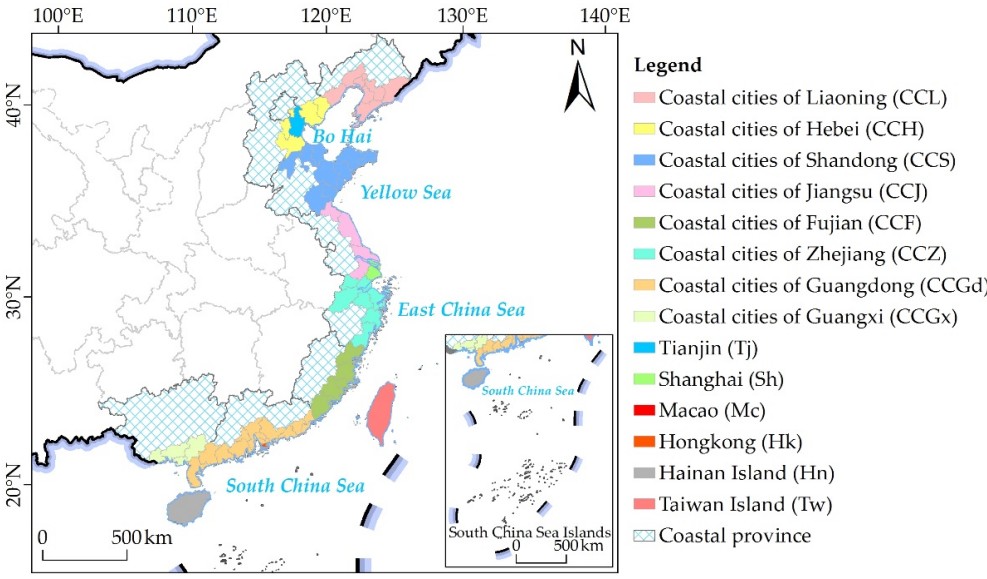

**Figure 1.** Study area.

*2.3. Methods*

2.3.1. Urban Expansion Quantification

Three indicators were calculated to quantify the magnitude of urban expansion: annual increase (AI, km² a⁻¹), new urban land proportion (NP, %), and annual growth rate (AGR, %). AI and NP were used to measure the change in urban area for a single city over different periods, and AGR was used to compare the urban expansion of a number of cities for a certain period while discounting the impact of city size [22,23,31]. The calculation formulas are as follows:

$$\text{AI} = \frac{U_{end} - U_{start}}{n} \tag{1}$$

$$\text{NP} = \frac{A_{t_i}}{\sum_1^n A_{t_i}} \tag{2}$$

$$\text{AGR} = \left[ \left( \frac{U_{end}}{U_{start}} \right)^{1/n} - 1 \right] \times 100\% \tag{3}$$

where $U_{start}$ and $U_{end}$ are the urban area extents at the start and end of a certain period, respectively, $n$ is the number of years spanned, $A_{t_i}$ is the area of new urban land in period $t_i$, $i$ is the number of periods.

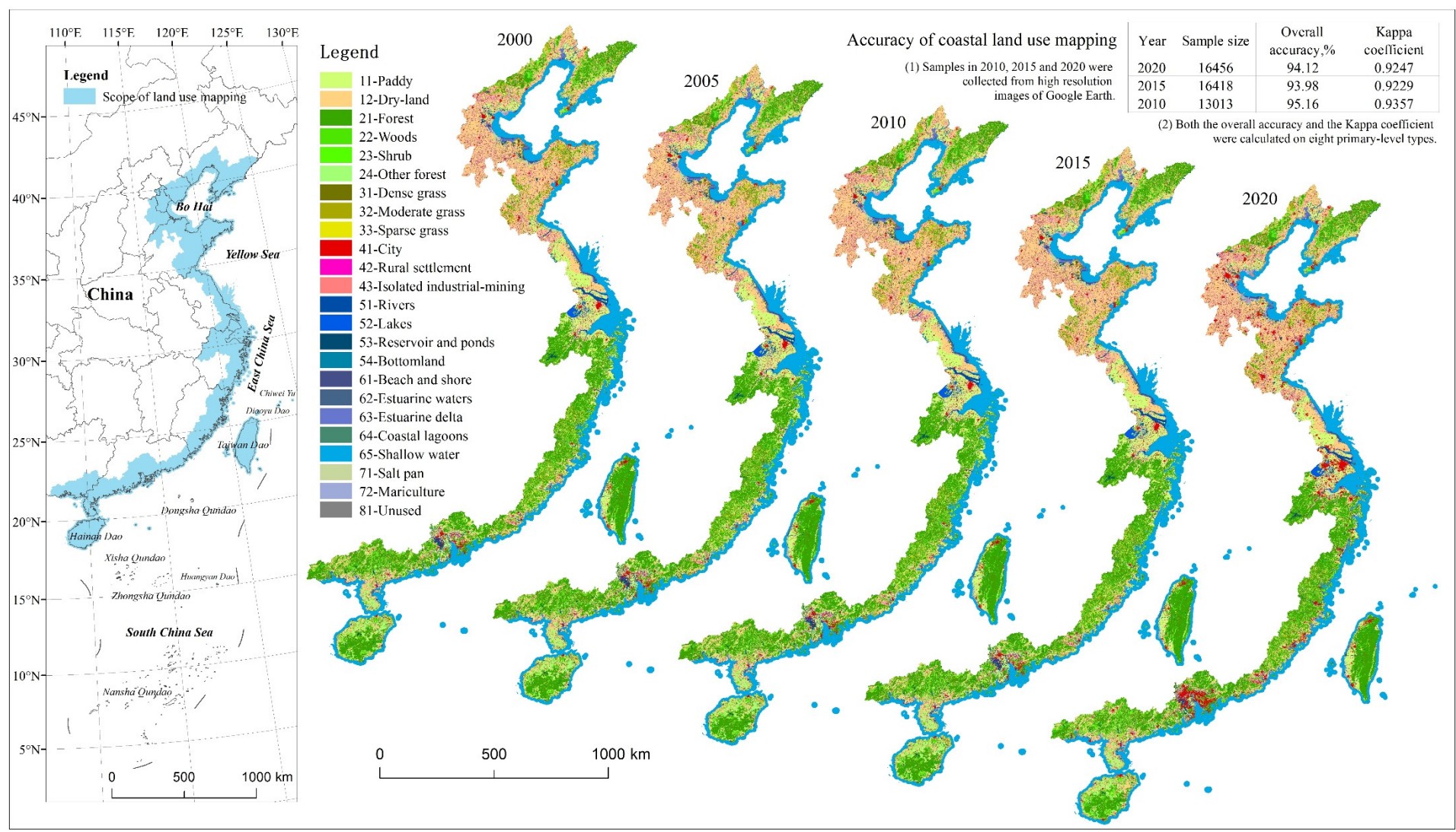

**Figure 2.** Land use maps in China's coastal area based on remote sensing techniques (obtained by updating Figure 1 in [47]).

### 2.3.2. Urban Expansion Types

According to the positional relationship between new urban land and the original urban land, urban expansion types are mainly divided into three types: infilling, edge-expansion, and outlying [24,38]. In this paper, the landscape expansion index (*LEI*) was used to identify different types of urban expansion. The *LEI* is calculated as follows:

$$LEI = \frac{S}{S + S_v} \times 100 \tag{4}$$

where *S* refers to the area occupied by original land in the buffer zone for new urban land, $S_v$ refers to the area of the vacant category in the buffer zone. Following Wu et al. (2015) [22] and Yang et al. (2019) [26], the buffer distance was set to 1 m in this paper. Urban expansion type is defined as outlying, infilling, and edge-expansion when $LEI = 0$, $LEI > 50$ and $0 < LEI \leq 50$, respectively.

For a particular city, the mean landscape expansion index can be used to define the degree of urban expansion aggregation [48]. The MLEI is calculated as follows:

$$\text{MLEI} = \sum_{i=1}^{n} \frac{LEI_i}{n} \tag{5}$$

where $LEI_i$ is the *LEI* of new urban land *i*, *n* is the amount of new urban land. Generally, the larger the value of MLEI, the more concentrated the distribution of new urban land is.

## 3. Results

### 3.1. Quantification Characteristics of Urban Expansion in China's Coastal Zone during 2000–2020

From 2000 to 2020, China's coastal zone experienced rapid urbanization, especially cities in the Bohai Rim, Shandong Peninsula, the Yangtze River delta, and the Pearl River delta (Figures 2 and 3). The urban extent of China's coastal zone increased from 11,473.88 km² in 2000 to 29,453.60 km² in 2020, with an annual increase (AI) and annual urban growth rate (AGR) of 898.99 km² a⁻¹ and 4.83%, respectively.

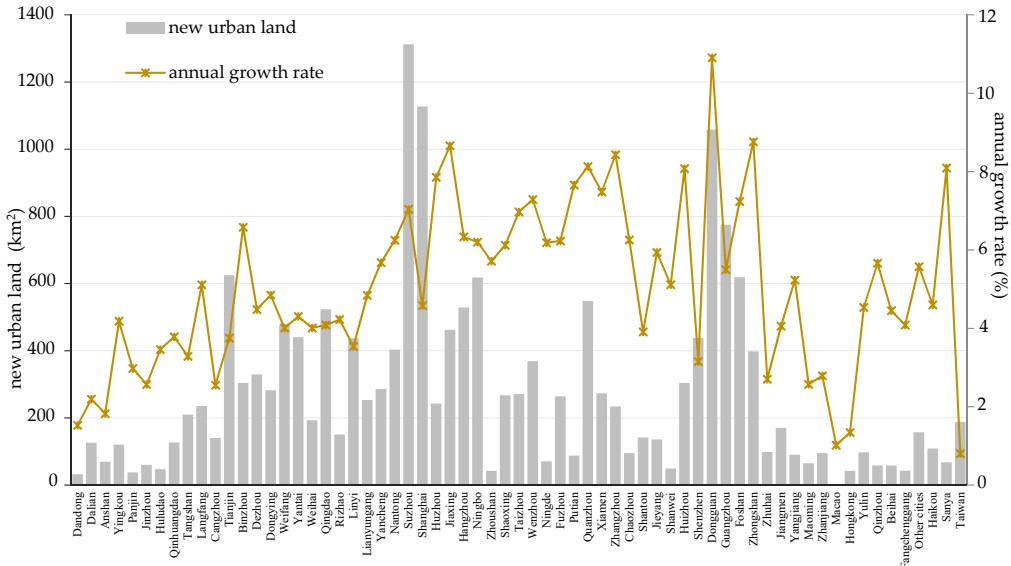

**Figure 3.** New urban land and annual growth rate for cities in China's coastal zone during 2000–2020. Notes: The order of the cities from left to right in the horizontal coordinates corresponds to the order of the geographical position of the cities from north to south in China's coastal zone, except for Taiwan as last; The vertical coordinates on the left and right represent the increase in urban land and the annual growth rate of urban land in coastal cities over the past 20 years, respectively.

At the regional level, for the coastal area in 14 provinces, municipalities and special administrative regions, the distribution of new urban land showed significantly spatial differences. For example, the area of new urban land in CCGd was 4532.45 km$^2$, which accounted for 25.21% of total new urban land in China's coastal zone, and approached the total increases of CCL, CCH, and CCS (27.62%). At prefecture level, cities with a large increase in urban land were Suzhou, Shanghai, and Dongguan; the areas of new urban land were 1312.24 km$^2$, 1127.21 km$^2$, and 1058.30 km$^2$, with corresponding annual growth rates of 7.04%, 4.58%, and 10.91%, respectively. The cities with small increases in urban land were mostly concentrated in CCL, CCH, CCF, Mc, Hk, and Tw, with new urban land and annual growth rates under 200 km$^2$ and 4%, respectively. Some cities with small increases in urban land had high annual growth rates; for example, the areas of new urban land in Qinzhou and Sanya were 58.21 km$^2$ and 67.91 km$^2$, with corresponding annual growth rates of 5.66% and 8.09%, respectively.

With 5-year intervals, the urbanization process of China's coastal zone was divided into four periods: 2000–2005, 2005–2010, 2010–2015, and 2015–2020, which basically corresponded to China's five-year plans on national economic and social development. During the four periods (as shown in Table 1), the areas of new urban land in China's coastal zone were 5052.35 km$^2$, 5074.46 km$^2$, 2533.52 km$^2$, and 5319.39 km$^2$, respectively, with a sharp decline in 2010–2015. Among the four periods, a significantly decreased trend for AGR was observed before 2015 with values of 7.57%, 5.50%, and 2.24% in the corresponding periods of 2000–2005, 2005–2010, and 2010–2015, while the values of MLEI exhibited an opposite trend before 2015. During 2015–2020, the new urban land and AGR values increased, while the MLEI values decreased distinctly.

**Table 1.** Indicators of urban expansion in China's coastal zone during 2000–2020.

| Indicators | 2000–2005 | 2005–2010 | 2010–2015 | 2015–2020 | 2000–2020 |
|---|---|---|---|---|---|
| new urban land(km$^2$) | 5052.35 | 5074.46 | 2533.52 | 5319.39 | 17,979.72 |
| AI (km$^2$ a$^{-1}$) | 1010.47 | 1014.89 | 506.70 | 1063.88 | 898.99 |
| AGR (%) | 7.57 | 5.50 | 2.24 | 4.06 | 4.83 |
| MLEI | 31.92 | 49.69 | 54.00 | 41.38 | 37.57 |

*3.2. Spatiotemporal Difference of Urban Expansion during 2000–2020*

3.2.1. Spatiotemporal Characteristics of New Urban Land

New urban land proportions (NP), annual increase (AI), and annual growth rate (AGR) of coastal cities were derived using Equations (1)–(3) to illustrate the spatiotemporal differences of urban expansion in four periods (Figure 4).

As shown in Figure 4, CCL, CCH, CCS, CCZ, and CCF experienced rapid urban expansion and had a significant increase in urban land area before 2015, while CCJ, CCGx, and Hn showed rapid urbanization after 2015. To highlight the differences in the magnitude of urban expansion among coastal cities, graded maps of AI and AGR for each city in different periods are shown in Figure 5.

Compared to 2000–2005, an increasing trend of NP, AI, and AGR occurred in CCL, CCJ, Sh, CCF, and Mc during 2005–2010, especially CCL and CCF. The values of NP for Dalian, Yingkou, Jinzhou, and Xiamen increased significantly, with corresponding values of 57.46%, 65.90%, 51.99%, and 65.31%, representing 2.90, 4.72, 9.06, and 3.82 times the NP values in 2000–2005. In addition, some cities in CCH, CCS, CCZ, and CCGd showed a slight decrease in NP, AI, and AGR compared to 2000–2005, such as Tangshan, Dezhou, Jiaxing, Hangzhou, Ningbo, and Zhongshan.

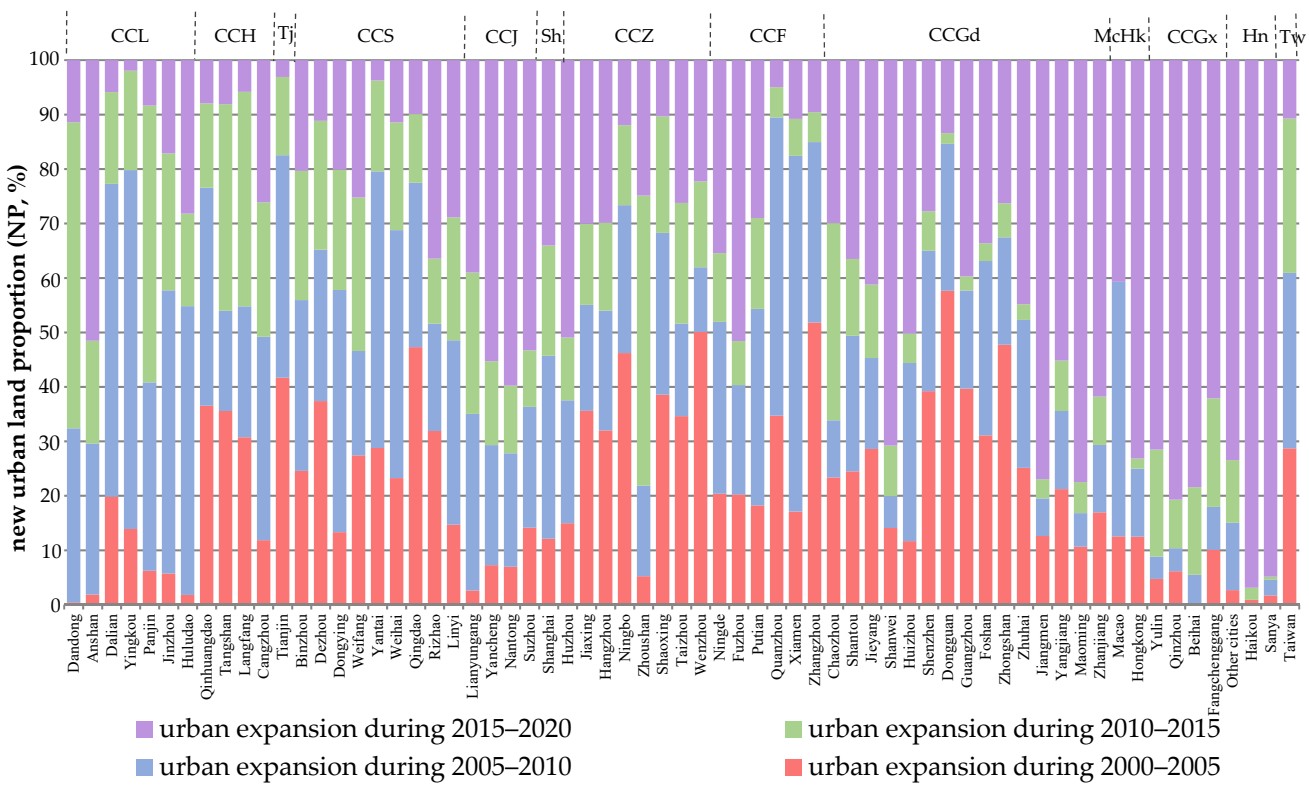

**Figure 4.** NP values of coastal cities in different periods. Notes: The order of the cities in the horizontal coordinates is consistent with that of Figure 3. During 2000 to 2005, coastal cities in the Bohai Rim, Shandong Peninsula, Yangtze River Delta, and Pearl River Delta regions had a rapid urbanization process, with NP, AI, and AGR at a high level among the four periods. Specifically, the maximum values of NP, AI, and AGR were observed in Dongguan, with 57.64%, 122.01 km$^2$ a$^{-1}$, and 37.93%, respectively. However, the urbanization process of coastal cities in other regions was slower, with values of NP below 20%. For cities with AI values below 1 km$^2$ per year, the values of AGR were generally below 1%, while AGR exhibited higher values in Zhoushan, Qinzhou, and Sanya, with values of 2.04%, 2.35%, and 1.25%, respectively, corresponding to their urban extents of 20.86 km$^2$, 28.96 km$^2$, and 18.15 km$^2$ in 2000.

During 2010–2015, the NP, AI, and AGR values of most cities in CCS, CCJ, CCF, and CCGd had decreased to different degrees compared with the previous two periods, with CCF and CCGd especially showing larger decreases; for example, the area of new urban land in CCGd was less than one quarter that in 2005–2010. Though most coastal cities in CCL, CCH, and CCS exhibited a decreasing trend compared to 2005–2010, their NP values were still at high levels compared to cities in other regions. For cities including Dandong, Panjin, Tangshan, Langfang, Weifang, Zhoushan, Taizhou, Wenzhou, and Chaozhou, as well as CCGx, their values of NP, AI, and AGR increased. In particular, the NP value of Zhoushan reached 53.28% with the AGR value at 11.86%.

During 2015–2020, the cities with higher NP, AI, and AGR values were concentrated in the Yangtze River Delta, northern CCGd, CCGx, and Hn, with a significant increasing trend compared to other periods, especially for CCGx and Hn which experienced rapid urbanization, with all the NP and AGR values greater than 60% and 8%, respectively. For example, the NP, AI, and AGR values of Sanya were 94.85%, 12.88 km$^2$ a$^{-1}$, and 31.78%, respectively. However, CCL, CCH, Tj, and CCS entered into slow urbanization, especially Yingkou, Tianjin, and Yantai, for which the values of NP were below 5%. In addition, Suzhou had the largest new urban land area, which even exceeded the total new urban lands of CCGx and Hn during the past 20 years, with NP, AI, and AGR values of 53.29%, 139.85 km$^2$ a$^{-1}$, and 10.62%, respectively.

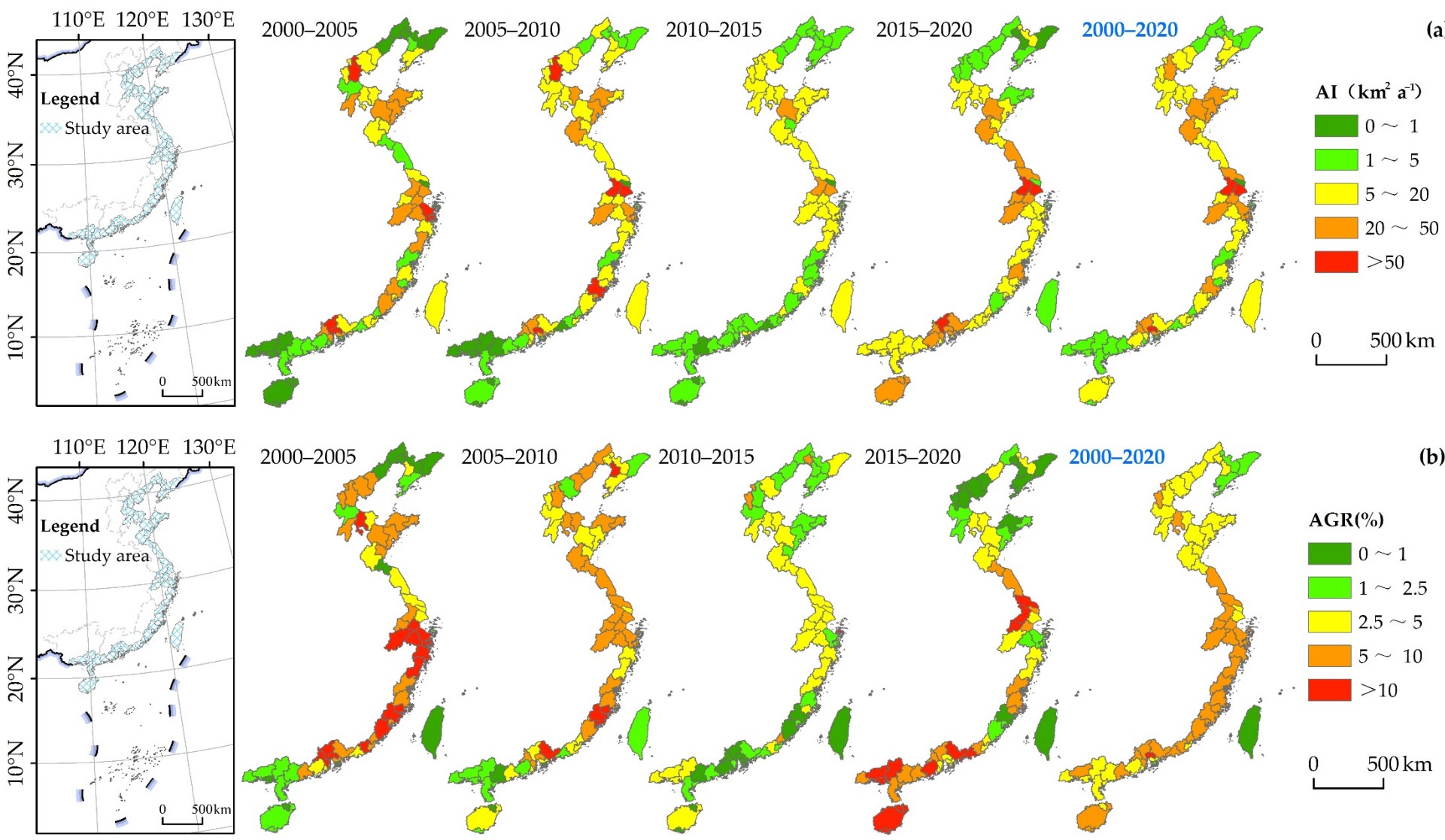

**Figure 5.** Quantification of urban expansion in China's coastal zone: (**a**) annual increase (AI); (**b**) annual growth rate (AGR).

3.2.2. Urban Expansion Types in Different Periods

Urban expansion was classified into three types using Equation (4), including edge-expansion, outlying, and infilling, respectively. Figure 6 shows the area proportions of the three types in coastal cities for the four periods, and Figure 7 shows cities exhibiting different expansion characters. Generally, the edge-expansion type was the main type of new urban patches in China's coastal zone over the past 20 years, while there were still significant spatiotemporal differences. For most coastal cities, with urban expansion, the area proportions of the edge-expansion type tended to decrease in the first three periods and then to increase during 2015–2020. The infilling and outlying types showed an opposite trend and fluctuating changes, respectively.

During 2000–2005, the urban expansion of China's coastal zone was dominated by edge-expansion, especially CCJ, Sh, CCZ, and CCGd, for which the proportions of edge-expansion were mostly over 70%. Cities with a relatively large proportion of the outlying type were concentrated in CCS, southern CCZ, and CCF, though some cities, such as Dongying and Weihai, were characterized by an outlying process.

During 2005–2010, edge-expansion was still the prevailing type of urban expansion with a slight decrease in most coastal cities but with a dramatic increase in the coastal cities of Shandong. The proportions of the infilling type in the cities of China's coastal zone generally increased compared to 2000–2005, while the trend of the outlying type was opposite, though showing a large proportion in a few cities of Guangdong.

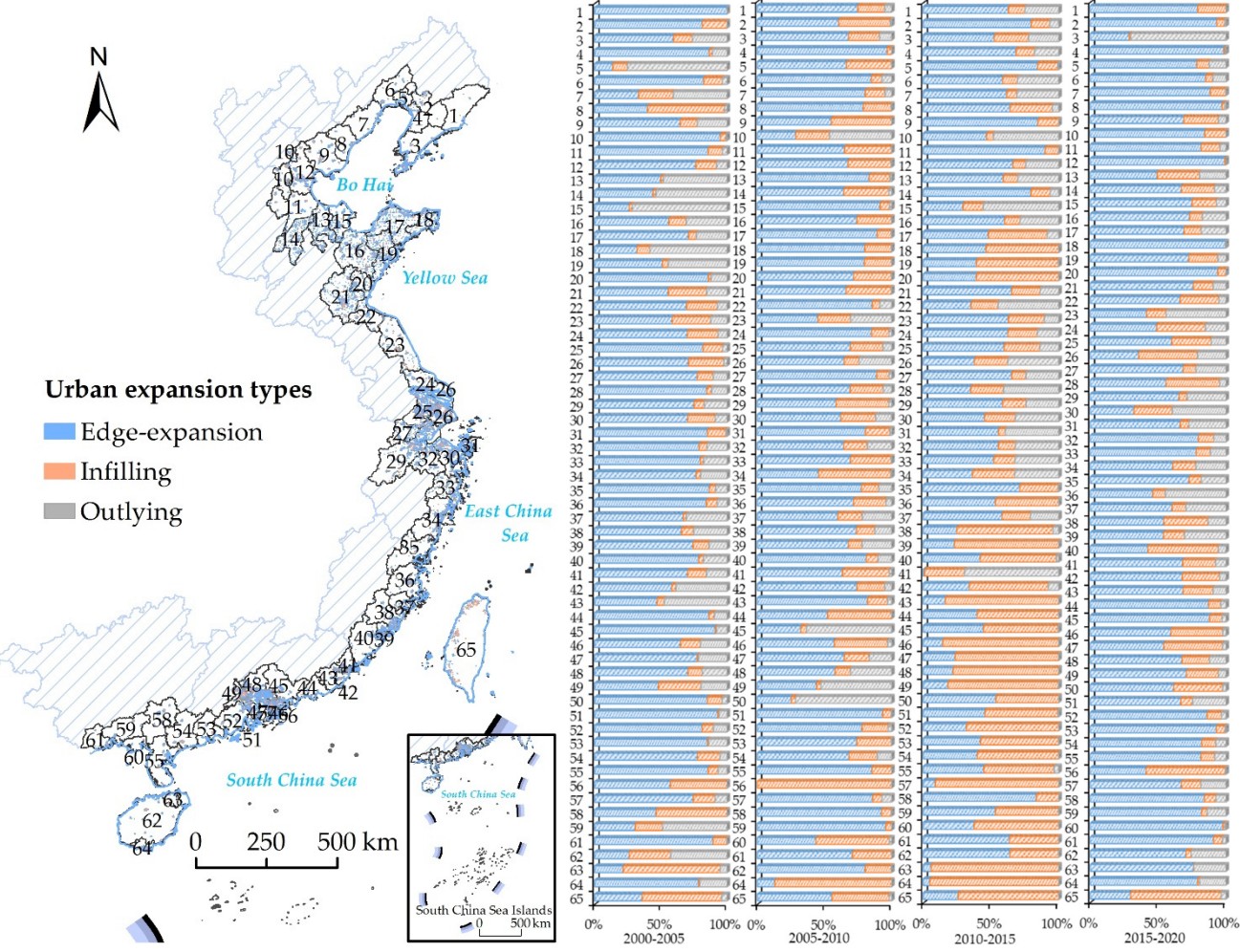

**Figure 6.** Proportions of three urban expansion types in coastal cities during 2000–2020.

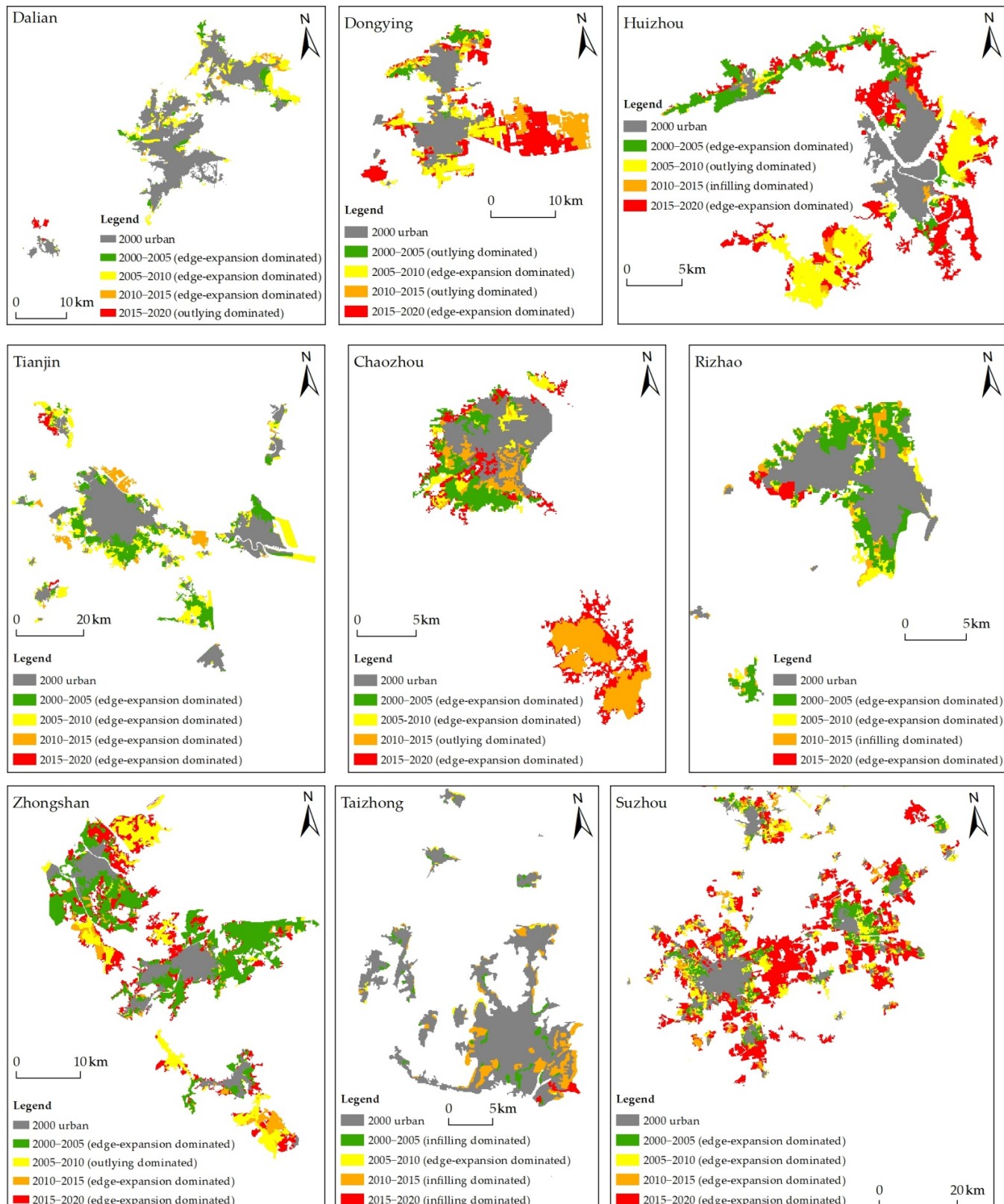

**Figure 7.** Spatio-temporal characteristics of urban land in typical cities during 2000–2020.

During 2010–2015, the dominant expansion type varied significantly. For cities in southern CCF, CCGd, CCGx, Hn, and Tw, the dominant type gradually shifted from edge-

expansion to infilling, except in Chaozhou which was dominated by the outlying type with a proportion at 69.52%. For most of the other cities, the urban expansion type still prevailed over edge-expansion, except for Dongying (as shown in Figures 6 and 7). In addition, in cities in which the edge-expansion type predominated, especially cities in CCZ, the proportions of the outlying type increased to various degrees compared with the previous 5-year period.

During 2015–2020, the prevailing type of urban expansion returned to edge-expansion in most coastal cities of China, and the proportions of the infilling type decreased significantly. In addition, the outlying patches were dispersed all over the coastal zone in relatively small proportions.

As shown in Figure 8, excluding CCH and Mc which showed fluctuating MLEI values during 2000–2020, the MLEI values of coastal cities experienced a significant increase before 2015, particularly in the third period (2010–2015). The values of MLEI in CCGd were higher than for other cities or other three 5-year periods, while the MLEI values in most of CCL, CCH, and CCS experienced a similar trend, increasing and then decreasing before and after 2010, respectively. In the last 5-year period (2015–2020), the MLEI values of most coastal cities decreased to various degrees, while the values continued to increase in Sh and CCJ.

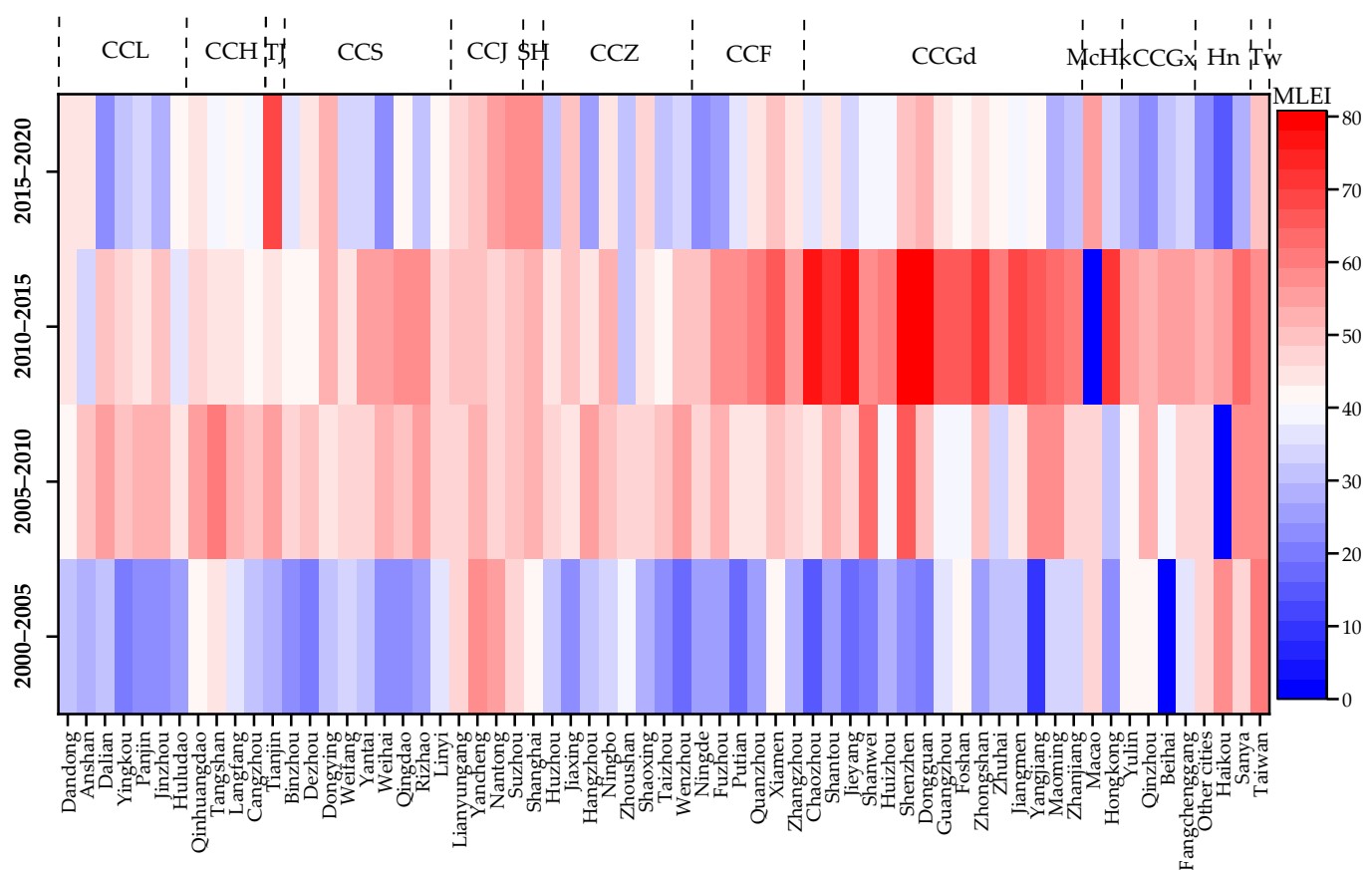

**Figure 8.** MLEI values of coastal cities in different periods. Notes: The order of the cities in the horizontal coordinates is consistent with that of Figure 3.

## 4. Discussion

### 4.1. Spatiotemporal Dynamics of Urban Expansion in China's Coastal Zone

Time series land use data, and multi-temporal analysis methods combined with indictors representing urban expansion patterns and types can effectively reflect the dynamic process of urbanization in terms of urban land [17–21,49]. A large number of studies have considered the spatial and temporal characteristics of urban expansion in typical cities or urban agglomerations of China based on remote sensing techniques [16,20,22,23,25,30–32,34,50],

which have demonstrated that China's urban expansion has significant regional difference. However, studies on typical cities, such as provincial capitals [30] or urban agglomerations [50], might ignore the characteristics of small cities or cities not radiated by urban agglomerations; thus, it is necessary to conduct a comprehensive analysis for a wide range of regions with cities of different sizes. This study provided a new perspective on the whole coastal zone rather than a local area or a single city in a coastal zone. Overall, the urban extent of China's coastal zone has expanded significantly during the past 20 years, and the speed of urbanization in coastal cities has exhibited prominent spatial-temporal differences.

During 2000–2020, the magnitude and type of urban expansion showed obvious spatiotemporal differences at city and regional level, and there were significant temporal differences in the urbanization process. Spatially, new urban land was mainly concentrated in economically developed regions, with notable differences among cities. Temporally, the urbanization process was unstable. For CCL, CCH, CCS, Southeast CCF, and Tw, the urbanization rate decreased, with the magnitude of urban expansion sharply declining in the last 5-year period (2015–2020). Meanwhile, in the south of China's coastal zone, such as CCJ, CCGx, and Hn, there was slow urbanization before 2015, and new urban land increased dramatically in 2015–2020.

In addition, urban expansion in China's coastal zone was mainly dominated by edge-expansion after 2000, and the area proportions of infilling and outlying varied significantly during the four five-year periods. In particular, during 2010–2015, the dominant expansion type shifted from edge-expansion to infilling in most cities of CCGd, CCGx, and Hn. Meanwhile, the increase in urban land in China' coastal zone was much lower than that during the other three 5-year periods—this was most probably related to the global financial crisis of 2008. Under the impact of the global financial crisis, economic activities reduced, and urban expansion slowed down markedly, even in developed cities, such as Guangzhou and Shenzhen.

Previous studies have found that urban growth rate is strongly related to city size [34,35], while other studies have claimed that urban growth rate is independent of city size, which is expressed in Gibrat's law [51–53]. However, this study found that urban growth rates of small cities could be higher than that of large cities in certain periods. For example, the urban growth rates of CCGx and Hn were higher than other cities in the last period (2015–2020) (Figure 4), but the original urban extents in 2015 of these cities were much smaller than others. Considering Sanya and Suzhou, in 2015–2020, the annual urban growth rate of Sanya was three times that of Suzhou, while the annual increase in urban land was less than one tenth of Suzhou's. This phenomenon might be related to the regional development strategies put forward in different periods.

### 4.2. Impacts of Policies on Urban Expansion

Major policies, such as economic development strategies and regional development plans, are significant drivers of the dynamic evolution of urban expansion [22,23,37]. In this paper, the results indicated that there were obvious spatial and temporal differences in urban expansion in China's coastal zone during 2000–2020, and such differences had significant consistencies with major policies, such as the Five-year Plan (Outline of the Five-Year Plan for Social and Economic Development), special economic zones, city clusters, revitalization of the northeast, and so on.

Figure 9 shows the national and regional development plans implemented in China's coastal zone since 2000. After China's accession to the World Trade Organization (WTO), China's economic development began a new stage, with regions involving the Bohai Rim, Shandong Peninsula, Yangtze River Delta, Southeast Fujian and Pearl River Delta entering a rapid stage of urbanization, and urban extent increasing significantly (Figures 3–5). For example, as a special economic zone with resource and development advantages, Dongguan had the largest annual growth rate (AGR) in 2000–2005. However, with the subprime crisis spreading all over the world in 2008, causing the global stock market shock and economic recession [54,55], economic activities reduced in Dongguan, which led to

new urban land in Dongguan declining dramatically, especially in 2010–2015 when the area of new urban land was less than one thirtieth that in 2000–2005. In order to promote the economic development of Guangdong, a series of national and regional development plans were proposed after 2008, such as the Pearl River Delta Industrial Layout Integration Plan implemented in 2009. With a series of policy developments, promoting economic recovery, the area of new urban land of CCGd increased significantly during 2015–2020 (Figures 3 and 4).

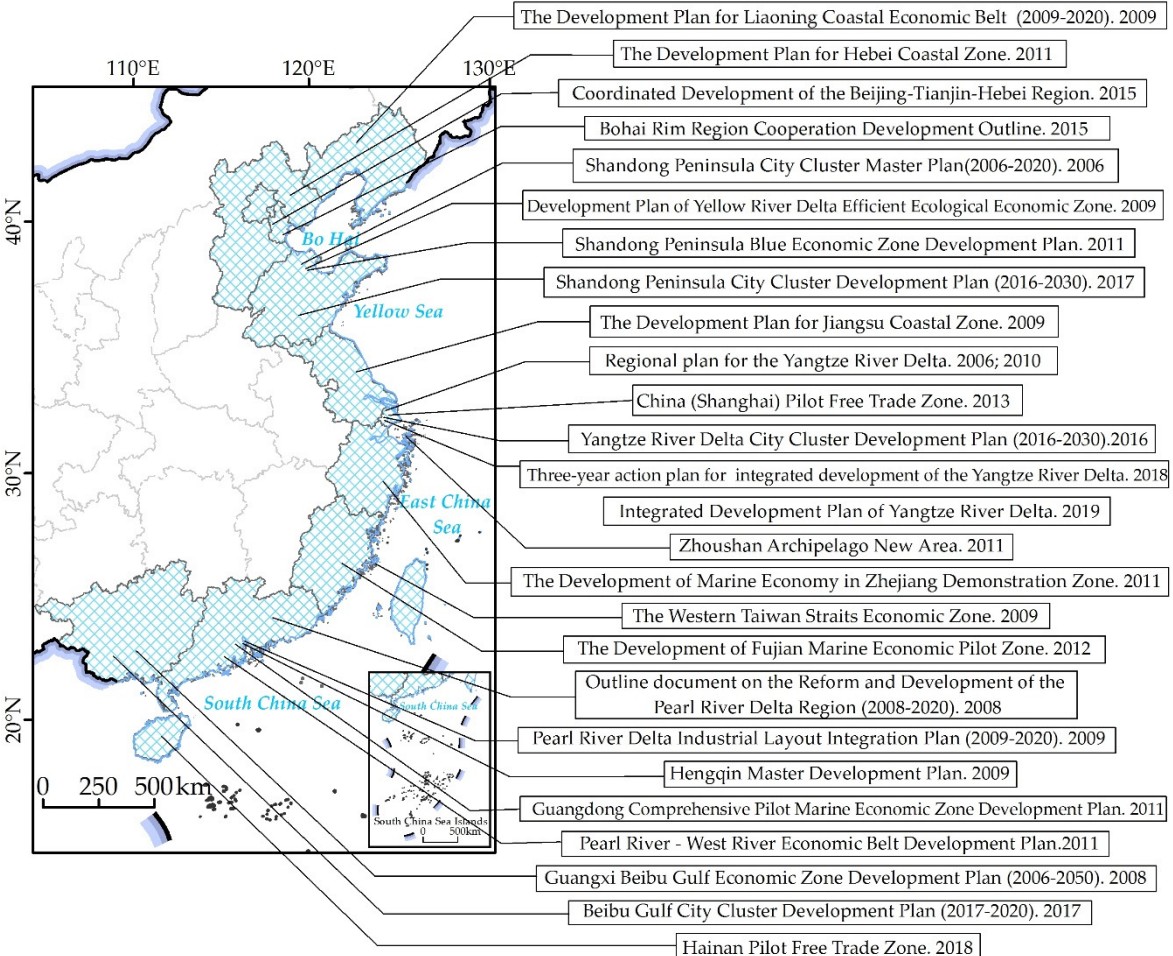

**Figure 9.** National and regional development plans in China's coastal zone.

As problems, such as environmental pollution, reduced ecological diversity, and land abuse, have intensified, governments have paid more attention to the sustainable utilization of resources, and more and more sustainable development strategies concerning land intensification and environmental protection have been proposed, such as the cultivated land protection policy. During the three early periods (2000–2005, 2005–2010, and 2010–2015), with the development concepts of intensive development and regional economic integration put forward, the values of MLEI increased significantly, and the proportions of urban land infilling increased. This indicated that the development strategies strongly promoted urban expansion change from extensive to intensive [30,40].

In addition, as the revitalization of the northeast strategy proposed in 2004, CCL experienced rapid urban expansion at an incredible rate after 2005, and the NP, AI, and AGR values increased significantly. This demonstrated that regional development strategy had impacts on the urban expansion in different periods.

Moreover, as a new territorial unit for national participation in global competition, urban agglomerations have gradually become the main form to achieve regional economic

integration and to drive the development of neighboring cities by radiation effects [32,56,57]. The scale, urbanization level, and agglomeration of city clusters had huge variability and unevenness [38,56–58], which manifested in significant spatial differences and stage differences in the urbanization process in China's coastal zone. Driven by the radiation effects of city clusters, CCJ joined the Yangtze River Delta city cluster [59] in 2016, while CCGx and Hn joined the Beibu Gulf city cluster [60] in 2017, which accelerated urban expansion in those areas. The results showed that there were significant increases in the NP, AI, and AGR values in the cities of Jiangsu, Guangxi and Hainan during the fourth five-year period (2015–2020), and new urban land of CCJ, CCGx, and Hn was 1.13, 1.80, and 4.86 times greater than the sum of that during the previous three five-year periods combined (2000–2015), respectively; in particular, the new urban land in Sanya increased by 17.41 times compared to the other three five-year periods. The results also showed that the proportion of the outlying type in cities of China's coastal zone increased to different degrees during 2015–2020. This was mainly due to the radiation-driven effects of urban agglomerations, which had led to the expansion of new urban land patches to non-core areas of urban agglomerations and the increase in outlying patches.

*4.3. Current Limitations and the Way Forward*

In this paper, the proportion of new urban land in different periods, the annual increase in urban land and the annual urban growth rate were calculated to reveal spatiotemporal differences at regional and city levels. Theoretically, urban expansion includes not only the spread of urban land, but also the concentration of population, and there is a clear inconsistency between urban expansion and population expansion [61–64]. Many studies have found that population and economy play very important roles in urban land expansion [26,65,66]; therefore, it is of value to research the relationship among migration movements, economic growth, and urban land expansion. In addition, since rapid urban expansion results in the loss of farmland, forests and water, destruction of the ecological environment, and so on, it is important to analyze the ecological effects of urban expansion in China's coastal zone [13,57,58,67]. In future, the relationship of population expansion, economic growth, and urban land expansion as well as the ecological effects of urban expansion, should be considered to further reflect the characteristics of urbanization, and to help policymakers balance the relationships among land, population, environment, and development.

**5. Conclusions**

In this study, the dynamic evolution of urban expansion in China's coastal zone during 2000–2020 was analyzed based on LUCC data. The results showed that China's coastal zone experienced drastic urbanization during 2000–2020, with an average urban growth rate (AGR) of 4.83%. Cities with large new urban land area were concentrated in the regions of the Bohai Rim, Shandong Peninsula, Yangtze River Delta and Pearl River Delta, and the variation and trends of NP, AI and AGR in different regions showed dramatic spatiotemporal differences. In addition, the values of MLEI showed incredible increasing trends before 2015, which indicated that the new urban patches gradually became more concentrated over time. The temporal differences in urban expansion confirmed that the global financial crisis of 2008 had severely impacted the process of urbanization in China's coastal zone. Furthermore, the high consistencies between development policies and urbanization showed that national and regional development strategies and plans had probably played a very important role with respect to the differences in urban expansion in different cities or regions. Furthermore, driven by economic development policies, cities originally with a small area of urban land would be likely to exhibit a higher speed of urban expansion than many large cities.

In brief, it is worth noting that even in similar geographic environments, there are still significant differences in the development of coastal cities, and such differences are strongly related to the economic status and size of the city. As a result of policy guidance,

there are also significant differences with respect to the ability of cities to agglomerate population and to tap the potential for economic development, leading to unbalanced characteristics of urban development. In the future, it is important to study the relationship among population, policy, economy development, and urban expansion. This study will provide essential information to policymakers, governments and investors for formulating strategies or investing capital to improve the efficiency and quality of urbanization.

**Author Contributions:** Conceptualization, P.D. and X.H.; methodology, P.D. and X.H.; software and formal analysis, P.D., X.H. and H.X.; validation and outcome investigation, P.D.; writing—original draft preparation, P.D.; writing—review and editing, P.D., X.H. and H.X.; supervision, X.H.; project administration, X.H.; funding acquisition, X.H. All authors have read and agreed to the published version of the manuscript.

**Funding:** This work was supported by the Strategic Priority Research Program of the Chinese Academy of Sciences (grant No. XDA19060205) and the National Natural Science Foundation of China (grant No. 42176221).

**Institutional Review Board Statement:** Not applicable.

**Informed Consent Statement:** Not applicable.

**Data Availability Statement:** The data presented in this study are available on request from the corresponding author.

**Acknowledgments:** We would like to express thanks for the constructive comments from the editor and anonymous referees.

**Conflicts of Interest:** The authors declare no conflict of interest.

## Abbreviations

| | |
|---|---|
| CCL | Coastal cities of Liaoning |
| CCH | Coastal cities of Hebei |
| CCS | Coastal cities of Shandong |
| CCJ | Coastal cities of Jiangsu |
| CCZ | Coastal cities of Zhejiang |
| CCF | Coastal cities of Fujian |
| CCGd | Coastal cities of Guangdong |
| CCGx | Coastal cities of Guangxi |
| Tj | Tianjin |
| Sh | Shanghai |
| Mc | Macao |
| Hk | Hongkong |
| Hn | Hainan Island |
| Tw | Taiwan Island |
| NP | New urban land proportion |
| AI | Annual increase |
| AGR | Annual growth rate |
| LEI | Landscape expansion index |
| MLEI | Mean landscape expansion index |

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
