# Peer review of "Dynamic Expansion of Urban Land in China’s Coastal Zone since 2000"

_remotesensing, doi:10.3390/rs14040916_

Round 1

Reviewer 1 Report

Dear Authors, here I can see a normal paper that can be improved substantially for publication on Remote Sensing.

I give you some suggestions in the following lines:

  • There is a distinct lack of Remote Sensing perspective in various sections, throughout. In a journal like Remote sensing, it is one of the essential thing

A better justification regarding spatiotemporal analysis is required.:

2) The relationship between policy and economics needs to be investigated further in terms of the spatiotemporal and statistical aspect.

3) For a better comprehension, (Fig.6) should be revised. (Categorizing like Fig.3).

4) Some aspects of language usage can be improved (e.g., use of the word obviously is extensive).

Reviewer 2 Report

  1. General: This is an interesting study based on published data in an intensively investigated and published domain. The study helps to establish a typology and a classification of Chinese coastal cities. It fails in establishing a sustainable development context. In particular how socio-economic drivers explain the observed urban dynamics is not addressed. Also the environmental drivers and effects are marginally considered.
  2. Specific:                                                                                                       2.1 The introduction puts emphasis on the methodological aspects of the problem. The last paragraph defines clear targets.                             2.2 M&M: Land use data: The paragraph is vague. More detailled information is required so that an interested reader is (at least in principle) able to repeat what the authors did.                                        2.3 Results: Figure 1 is unclear: What does the upper line in the figure show? Describe and comment on the results in the text.                          Figure 3, figure 5 and figure 6 are most unclear. Redraw these figures using more contrast. Provide description in the text.                                2.4 Discussion:  In particular in China a lot of publications exist on related subjects. The discussion should also deal with similar/opposite results published in the open literature. Put emphasis on the originality of your research.                                                                                        2.5 The text uses quite a number of acromyms and abbreviations. It is advisable to add a list of acronyms and abbreviations.                             2.6 Although thehe text is clear and accessivbe, it is in need of an in depth language revision. In particular the style and the choice of words should be addressed.

Reviewer 3 Report

Dear Authors,

the idea of the paper is very interesting and can be considered for publication in a scientific journal. The analysis of urbanization effects are very interesting topics that are sometimes overlooked. The idea of quantified the spatiotemporal characteristics of urban expansion considering  the quantify and compare the urban extents, new urban increases ,urban growth rates, and urban expansion types in cities of  China’s coastal zone during different periods, analyze the spatiotemporal differences of urban expansion at city-level and region-level, discuss the consistencies between social-economic development policies and urban expansion in cities of China’s coastal zone is a good idea. 

However, I propose the following changes to improve the card:

  • Are there other studies from other countries dealing with the same problem? Deepen the state of the art, also looking for information in journals, questionnaires, statistical data, etc.
  • Discussion: I would increase the discussions to give more evidence to the 3 questions proposed in the introduction. Try to answer the 3 questions
  • Abstract: insert some hint of the results

Reviewer 4 Report

The manuscript measures the dynamic expansion of urban land in China’s coastal zone since 2000. The manuscript is easy to follow. 

  1. Line 15, "based on remote sensing techniques", the authors only used manually interpreted land use data, there are no remote sensing techniques.
  2.  What's the definition of urban land? And the difference between urban land and built-up area?
  3.  Line 95-98: Please be careful of the statement, I don't think the statement is correct.
  4. Line 129: Can you explain the definition of the 'city'.
  5.  Eq.2: What's the letter N represent? the same with letter n?
  6. Eq.3: Can you explain why use an exponential number of 1/n? Eq. 3 is hard for readers to interpret.
  7. Line 152-153: Can you explain the difference between 'infilling' and 'edge-expansion'?  Why do both of them have a LEI less than 50, how to distinguish them?
  8.  Conclusion section, please do not simply summarize the points already made in the body, try to interpret the findings at a higher level of abstraction. Please also include ideas of what could or should still be done in relation to the issue addressed in your paper.
  9.  Please also include a figure to show the land use data.

Round 2

Reviewer 1 Report

Accepted as it is

Reviewer 4 Report

The authors have addressed my concerns. Just a minor remark: Please improve the quality of low-resolution figures.